# A Gradient Descent Optimizer with auto-controlled large Learning Rates, dynamic Batch Sizes and without Momentum

## Abstract

We present a novel, fast gradient based momentum-free optimizer algorithm with dynamic learning rate and dynamic batch size. The main ideas are to exponentially adapt the learning rate $\alpha$ by situational awareness, mainly striving for orthogonal neighboring gradients, and to increase the batch size when the gradients become too noisy, leading to random walks rather than gradient descent. The method has a high success and fast convergence rate and relies only on few hyper-parameters, providing greater universality. It scales only linearly (of order $O(n)$) with dimension and is rotation invariant, thereby overcoming known limitations. The optimization method is termed ELRA (**E**xponential **L**earning **R**ate **A**daption). The impressive performance of ELRA is demonstrated by experiments on several benchmark data-sets (ranging from MNIST to ImageNet) against common optimizers such as Adam, Lion and SGD.

## 1 Introduction

Numerical optimization of functions $f(x)$ relies on data obtained from the function landscape. One key problem is that we are lacking meaningful global information about $f(x)$, making it necessary to rely on local information instead. Approaches based on local properties range from using the function value in physics-inspired relaxation approaches (cf. Borysenko & Byshkin (2021)), to algorithms using the topographical structure of the function landscape directly, such as gradient descent-like approaches, to biology inspired algorithms such as swarm optimization (cf. Ab Wahab et al. (2015)). Among these, the gradient descent-like methods have the longest history and are (due to their linear scaling with the problems dimension) the only practically applicable algorithms in high dimensional problems (e.g. deep neural networks). In these approaches the gradient $G = \nabla f(x)$ of the function $f(x)$ is computed and thus also the best descent direction $-G$. However, while the idea of going downhill is obviously reasonable, an optimal step-length $\lambda = \alpha \cdot ||G||$ remains to be chosen cost-efficiently. The parameter $\alpha$ is the learning rate (or step size). Most current gradient based algorithms use a fixed learning rate $\alpha$, which additionally may depend on time/steps $t$. This holds in particular for the Ada-family[1] of optimizers, widely used for training neural networks. To eliminate the initial tuning of $\alpha$, there are some modern approaches which adapt $\alpha$ dynamically, such as AdaDelta or Prodigy (proposed in Mishchenko & Defazio (2023)) or DoG (proposed in Ivgi et al. (2023)). Yet, they perform not better then the most prominent Ada-optimizer Adam (cf. Kingma & Ba (2017)) or its successful predecessor Lion (cf. Chen et al. (2023)).

The use of a fixed learning rate $\alpha$ is in part due to the fact that it allows, at least locally, for precise mathematical analysis, guaranteeing or almost surely guaranteeing (for SGD) a lower bound on convergence rates (e.g. see Nesterov (2018), 1.2.3). However, these lower bounds often depend on strong assumptions (such as convexity) and on constants which are in practice unknown. Moreover, more complex optimizers, such as Adam, even tend to fail general convergence (cf. Bock & Weiß (2019)), although they perform very reliable in practice.

We propose a paradigm changing algorithm that estimates in each step ***self-consistently a near optimal*** learning rate $\alpha$ from low-cost local knowledge of the function, thereby achieving a jump

---

[1]Such as: AdaGrad, RMSProp, AdaDelta, Adam, Lion, which all scale the gradient components individually (precondition-like).

close to the next minimum along the gradient direction. In particular, $\alpha$ approaches a problem-specific good scale *exponentially fast* (cf. Fig. 2a) and $\alpha$ is continually updated. We propose **ELRA** – **E**xponential **L**earning **R**ate **A**daption as a name for the new optimizer based on this idea.

Recent articles indicate that large variations of $\alpha$ might be very beneficial. In Grimmer (2023), it is for the first time mathematically proven that (periodically) varying step sizes lead to much better convergence rates, which our experimental results confirm. In Truong & Nguyen (2021) it is shown that estimating the best $\alpha$ via backtracking using Armijo's condition (see Nesterov (2018), 1.2.3) can lead to faster convergence then the Ada-family. However, each backtracking step needs a separate and expensive function value. Hence, backtracking more than once is seldom justified by the speed gained. ELRA does not suffer from this computational conundrum, as we provide a low-cost estimator (see sections 3.1 and 3.6) for the best $\alpha$, thereby retaining the benefit of a good $\alpha$ without losing speed.

The first essential advantage of ELRA is that a strongly adaptive $\alpha$ completely eliminates the need to find 'by hand' a good constant $\alpha$ for each specific problem. Secondly, most modern training schemes rely on decreasing $\alpha$ over time to achieve better test accuracy. Yet the best timing is a priori unknown and often determined by educated guesses. The strong performance of ELRA (cf. Tab. 3-5) shows that a strongly adaptive $\alpha$ needs no external timing. Thirdly, ELRA is invariant under orthogonal transformations of $x$, such as rotations, unlike the Ada-family (see Fig. 3 for different behaviour for rotated coordinates), which due to adaptively scaling each gradient component looses the invariance. The lack of such an invariance can cause problems in geometric optimization (cf. Ling et al. (2022)) as it can lead to unwanted biases and artifacts, it can negatively effect the generalization of the trained network (cf. Zhou et al. (2020)) and it can drastically reduce the speed near saddle points (cf. Fig. 3). In addition to the learning rate, we propose also dynamic adaption of the batch size (cf. 3.3) and provide a kind of soft restart (necessary, as big $\alpha$ can lead to temporary instability). Moreover, we present a technique that can improve the final result, which we call boosting.

We are convinced that each of these features on its own can, to a varying degree, be also beneficial for other types of optimizers, like the Ada-family (see App., table 6 for a summary of their properties).

## 2 THE IMPORTANCE OF ORTHOGONAL GRADIENTS

All gradient descent methods for minimizing functions $f(x)$ follow the update scheme

$$x_{t+1} = x_t - \alpha \cdot \big((1-\beta)G_t + \beta M_t\big), \tag{1}$$

where $G_t = \nabla f(x_t)$ is the gradient at $x_t$, $M_t$ the momentum and $\beta$ the ratio between $G_t$ and $M_t$. For the Ada-family, $\alpha$ is essentially constant while $G_t$ is not actually the gradient, but a component-wise modification, which is dynamically adapted. In general, the use of component-wise adaption of gradient and momentum leads to a dependency on the coordinate system and the speed of the algorithm depends heavily on the concrete representation of the data (see Fig. 3). Moreover, an essentially constant or time-variable $\alpha$ has to be chosen with care, either using past results or initial calibration runs. We provide a completely new approach which overcomes these problems. Firstly, our method requires no momentum, i.e. $\beta = 0$ for us, simplifying equation (1) to:

$$x_{t+1} = x_t - \alpha_t \cdot G_t. \tag{2}$$

With constant $\alpha$ this would be the trivial gradient descent. However, our $\alpha_t$ is highly dynamically adaptive. The main idea is to use the angle between the current and previous gradient $G_t$ and $G_{t-1}$ to determine the adaptation of $\alpha_t$. A short proof of why this is reasonable can be given as follows: We want to find $\alpha$, such that $x_t = x_{t-1} - \alpha G_{t-1}$ is a local minimizer to the differentiable[2] function $f$ near a point $x_{t-1}$ in the direction of $-G_{t-1}$. For that, we consider the function $h(\alpha) := f(x_{t-1} - \alpha \cdot G_{t-1}) = f(x_t)$, which is $f(x_t)$ at the next point $x_t$, depending on the learning rate $\alpha$. Differentiating $h$ with respect to $\alpha$ yields:

$$h'(\alpha) = \langle \nabla f(x_t), -G_{t-1} \rangle = -\langle G_t, G_{t-1} \rangle, \tag{3}$$

where $\langle a, b \rangle$ denotes the scalar product. Note that $h'(0) = -\langle G_{t-1}, G_{t-1} \rangle = -||G_{t-1}||^2$ is negative (with $||a|| = \sqrt{\langle a, a \rangle}$ being the euclidean norm). This means that $h$, and hence $f$, decreases for small $\alpha$. In fact, $h$ decreases until it reaches a critical point $\alpha_{min} > 0$, where we have $h'(\alpha_{min}) = 0 \Leftrightarrow 0 = \langle G_t, G_{t-1} \rangle$. If $h$ has at $\alpha_{min}$ an extremum, then it is necessarily a local minimum and thus also a minimum of $f$ in the direction of $-G_{t-1}$.

---

[2]See Math. suppl. (7), why even for non-differentiable activation functions (e.g. ReLU), $f$ can assumed to be smooth.

This gives the following conclusion: For the optimal learning rate $\alpha$, providing locally the smallest $f(x_t)$, the current and previous gradient $G_t$ and $G_{t-1}$ are orthogonal to each other, i.e. $\langle G_t, G_{t-1} \rangle = 0$. Moreover if $\langle G_t, G_{t-1} \rangle > 0$ then $\alpha$ has to be increased, while for $\langle G_t, G_{t-1} \rangle < 0$ it has to be decreased to give a better result. Figuratively speaking (cf. Fig. 1): If we see Zig-zag or anti-parallel steps we should decelerate, while for primarily parallel steps we should accelerate.

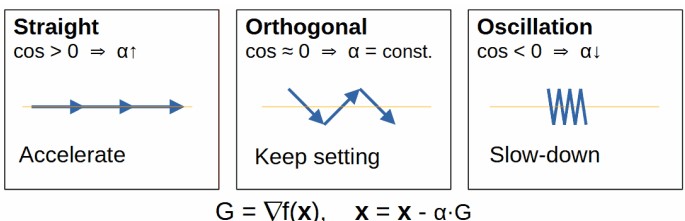

Figure 1: Situations during optimization and associated $\alpha$ updates.

As $\alpha_{min}$ depends continuously on $x_{t-1}$, we can expect that the optimal $\alpha_t$ for $x_t$ does not vary too much from the optimal $\alpha_{t-1}$ for $x_{t-1}$. This justifies the use of the scalar product $\langle G_t, G_{t-1} \rangle$ as an oracle for the next $\alpha_t$. Note that $\langle G_t, G_{t-1} \rangle$ is computational much cheaper than Armijo's condition (cf. Nesterov (2018), 1.2.3), as no extra gradient/function values are needed. Note that the above condition does not fix an $\alpha$-update-scheme. However, all feasable schemes can be written in the form $\alpha_t = \alpha_{t-1} \cdot (1 + \langle G_t, G_{t-1} \rangle \cdot g)$, where $g$ is any positive function.

## 3 THE ELRA OPTIMIZER

This section explains how the ELRA optimizer dynamically controls the learning rate $\alpha$ and the batch size $bs$. Furthermore we introduce the techniques of soft restarts and boosting. The code of the ELRA optimizer described here is online available via anonymous git (2024), using PyTorch.

### 3.1 THE $\alpha$-UPDATE FORMULA

In order to fix an explicit update formula for $\alpha_t$, we assume that the function $f$ is a parabola[3] along the straight line through $x_{t-1}$ and $x_t$, i.e. $f(x) = ax^2 + b$ in the direction $x_t - x_{t-1} = -\alpha_{t-1}G_{t-1}$. Note that here, $f$ is written using (in practice unknown) coordinates such that $x = 0$ is the minimizer of $f$. In this setting, the derivatives of $f$ are:

$$2ax_{t-1} = f'(x_{t-1}) = \partial_{G_{t-1}}f(x_{t-1}) = ||G_{t-1}||, \quad 2ax_t = f'(x_t) = \partial_{G_{t-1}}f(x_t) = \frac{\langle G_{t-1}, G_t \rangle}{||G_{t-1}||},$$

where $\partial_{G_{t-1}}$ is the directional derivative of $f$ with respect to $G_{t-1}$. Together with $x_t - x_{t-1} = -\alpha_{t-1}G_{t-1}$, we get the following update formula for $\alpha$, which is implemented in ELRA (see Methods, A.2, for a full derivation):

$$\alpha_t = \alpha_{t-1} \cdot \frac{||G_{t-1}||^2}{||G_{t-1}||^2 - \langle G_t, G_{t-1} \rangle} \cdot \kappa. \tag{4}$$

Here $\kappa \sim 1$ denotes an empirical correction term, which neutralizes random noise effects in neural networks. $\kappa$ is explicitly given as follows:

$$\kappa(\alpha_{t-1}) = 1 + 0.15 \cdot (1 + \alpha_{t-1}^2)^{-1}.$$

Note that the updated step size $\alpha_t$ can in principle be arbitrary between $-\infty$ and $+\infty$. We prevent this potentially catastrophic behaviour by imposing bounds of the form $0 < \alpha_t/\alpha_{t-1} < \gamma_{max}$, where $\gamma_{max}$[4] can be chosen at will, e.g. $\gamma_{max} \sim 10^6$. Moreover, we found that it is beneficial to impose the bounds $10^{-8} < \alpha < 10^6$ on $\alpha$. These are additional hyper-parameters, yet they are sufficient for all our experiments. For example for our CIFAR-10 experiments without weight-decay (see Results below), $\alpha$ ranges between $0.01$ and $10$. The initial $\alpha_0$ is another hyper-parameter. However, its choice is marginal, as ELRA adapts $\alpha_0$ exponentially fast (see Fig. 2a). We chose $\alpha_0 = 10^{-3}$ moderately to prevent initial instabilities of $f(x)$ (note the two lost runs for $\alpha_0 = 0.1$ and $1.0$ in Fig. 2a).

---

[3]A parabola ansatz is chosen, as near local minima, each function is almost a parabola (cf. Math. Suppl. C).
[4]For neural networks, $\gamma_{max} = 10$ is probably sufficient.

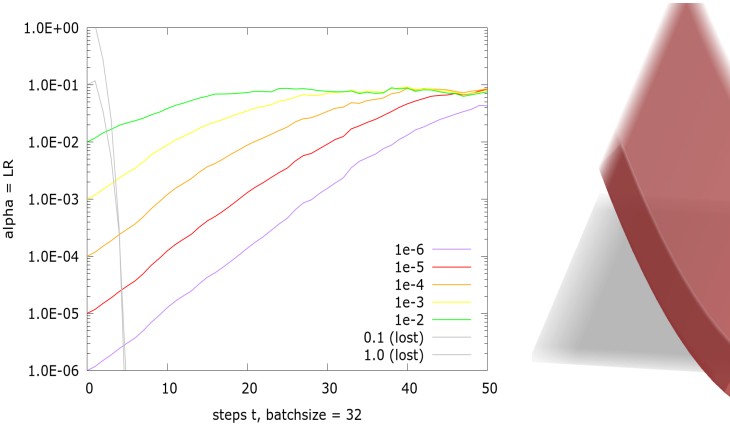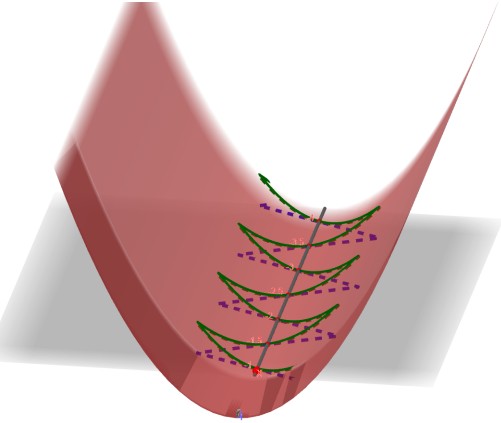

(a) Learning rates $\alpha$ in first 50 steps for CIFAR-10 for different initial $\alpha_0 = 10^{-6}, 10^{-5}, ..., 10^0$.

(b) Oscillating optimization path along a valley

Figure 2: Exponential adaption of learning rate $\alpha$ and oscillations along optimization path

For large language models such as BabyLlama, we need a warm-up phase for $\alpha$, where we impose on the step length $||G_t||\cdot\alpha_t \geq 0.1$ to prevent catastrophic $\alpha$-reduction[5]. Finally, we remark that our optimizer is by construction rotation invariant[6] (cf. Fig. 3), as it uses only scalar products. Moreover, we stress that computing scalar products is (relatively) cheap (of order $O(n)$ in time and space).

## 3.2 SOFT RESTARTS

The $\alpha$-update-scheme explained above can lead to numerical instabilities, due to overestimated increases for $\alpha$ (especially in almost linear parts of the landscape). To prevent a fatal increase of $f(x)$, we use retracing/soft restarts, if the new value $f(x_t)$ increases too much[7]. In these situations, we retrace back to the previous $x_{t-1}$ and update $\alpha_t$ by

$$\alpha_t = \alpha_{t-1} \cdot \frac{1}{2} \frac{\alpha_{t_1}||G_{t-1}||}{f(x_t) - f(x_{t-1}) + \alpha_{t-1}||G_{t-1}||}$$

a formula coming also from a parabola ansatz (using $f'(x_{t-1} = ||G_{t-1}||$ and the function values $f(x_{t-1})$ and $f(x_t)$) that decreases $\alpha$ at least by the factor $0.5$. The next point $x_{t+1}$ is then calculated by: $x_{t+1} = x_{t-1} - \alpha_t \cdot G_{t-1}$. The question when to retrace is delicate and leaves room for some individual choices. At the moment, we retrace if $f(x_t) > \overline{f(x)} + 5\sigma$, where $\overline{f(x)}$ is the average of $f(x_t)$ over the last epoch and $\sigma$ the corresponding standard deviation. The idea is that for $f(x_t) > \overline{f(x)} + 5\sigma$, the increase of $f(x_t)$ does almost surely not come from random noise but a bad choice of $\alpha$. Initially, when no average $\overline{f(x)}$ is yet known, we also retrace, if $f(x_t) > 1.1\cdot f(x_0)$, where $f(x_0)$ is the initial function value. This prevents fatal explosions of $f(x_t)$ in the beginning of the optimization. When starting at a pre-trained $x_0$, this initial condition can be dropped.

## 3.3 DYNAMICAL BATCH SIZE

Influenced by the article "Don't Decay the Learning Rate, Increase the Batch Size" by Smith et al. (2017), we also dynamically adapt the batch size $bs$. The picture behind this idea is the following: the gradients $G_t$ in neural networks carry a roughly fixed absolute amount of random noise. The optimization leads over time to smaller gradients (near the minimum), thereby increasing the relative noise until it leads to random walks rather than gradient descent, thus slowing or stopping the optimization. Increasing the batch size reduces the random noise, as $G_t$ is computed using more

---

[5]This is implemented by $\alpha_t = \max\{\alpha_t, 0.1/||G_t||\}$

[6]Actually even invariant under orthogonal transformations.

[7]Some increment of $f(x)$ is inevitable for stochastic gradients, as $f(x)$ might not decrease in the direction of $-G$.

training data, thus getting closer to the correct (noise-free) gradient computed using all data. However starting with a large $bs$ is also bad, as the computation time for gradients increases with larger $bs$, and using gradients with too little random noise can lead to overfitting by fast adaption to training data. We use a fixed minimal base batch size[8] $bs_{min}$ and obtain lager batches by accumulating integer multiples $m_t$ of minimal batches, i.e. $bs_t = m_t \cdot bs_{min}$. We update the accumulation number $m_t$ after a fixed number of processed training samples (often an epoch, 25% of an epoch for ImageNet and 45000 tokens for BabyLlama). Over this period, we compute the mean function value $\overline{f(x)}_s$ and its standard deviation $\sigma_s$ and we increase $m_t$ and hence $bs_t$ by a factor of 1.5 if either of the following three conditions hold:

$$(1)\ \overline{f(x)}_s > \overline{f(x)}_{s-1} + \frac{\sigma_s}{10}, \qquad (2)\ \frac{\sigma_s}{\overline{f(x)}_s} > \frac{1}{2}, \qquad (3)\ \overline{f(x)}_{s-3} < \overline{f(x)}_s, \overline{f(x)}_{s-1}, \overline{f(x)}_{s-2}.$$

We use condition (2) as a complement to (1), as it applies if the relative noise is large but does not lead to large increments of $f(x)$. Condition (3) triggers if $\overline{f(x)}$ does not get better within 3 control cycles. Actually, we suspect that (3) alone might suffice as a control mechanism.

To further reduce the danger of overfitting from fast adaption to training data, we increase $bs_t$ in cascades, meaning that every third update we decrease $bs_t$ by $1/1.5$ instead of increasing it.

### 3.4 THE GRADIENT-DECAY-FEATURE

By chance (a simple error), we found that the following gradient decaying feature can be helpful for datasets with strong overfitting (see results): When increasing the batch size by a factor of 1.5, we simultaneously decrease the gradient by 1.5, i.e. if the accumulated batch size is $bs_t = m_t \cdot bs_{min}$, we use the scaled gradients $\widetilde{G}_t = G_t \cdot \frac{m_0}{m_t}$. However, this feature is not always helpful!

### 3.5 MISCELLANEOUS

Firstly, consider the following often overlooked fact: The last batch of an epoch may contain much fewer elements than the other batches thus yielding a gradient that is noisier than the others. This can disrupt the optimization and give unreliable test losses (as they are calculated after the last batch). Especially optimizers using larger learning rates $\alpha$ and no momentum (such as ELRA) are effected by this phenomenon. Therefore, we always skip the last batch.

Secondly, as the norm of the gradient has a direct influence on the $\alpha$-update for ELRA, we cannot use gradient cropping to prevent numerical instabilities coming from very large single gradients. If necessary, we use step size cropping instead, i.e. we require $\alpha_t \cdot ||G_t|| \leq c$ for some constant $c$ (we use $c = 2$) and set $\alpha_t = \min\{\alpha_t, c/||G_t||\}$ to satisfy this condition.

### 3.6 EFFICIENT ARCHITECTURE

An important advantage of ELRA over most of the current optimizers is its momentum-freeness, as ELRA needs fewer vectors. As presented above, ELRA uses four vectors each step: the current point $x_t$, the current and last gradients $G_t$ and $G_{t-1}$ and (for retraces) the last point $x_{t-1}$. Moreover, only 3 vector operations are needed: the two scalar products $||G_{t-1}||^2$ and $\langle G_{t-1}, G_t \rangle$ and the subtraction $x_t - \alpha_t \cdot G_t$. This gives ELRA a roughly 10% computation advantage over Adam and Lion, when used with the same batch size. The memory requirements for ELRA could be further reduced as by

$$x_t = x_{t-1} - \alpha_{t-1} \cdot G_{t-1} \quad \Leftrightarrow \quad G_{t-1} = \frac{1}{\alpha_{t-1}}(x_{t-1} - x_t),$$

the last gradient can be recovered from the other three vectors. Moreover, we only use $G_{t-1}$ for its norm $||G_{t-1}||$, which can be calculated in the previous step, and for the scalar product

$$\langle G_{t-1}, G_t \rangle = \frac{1}{\alpha_{t-1}}\Big(\langle x_{t-1}, G_t \rangle - \langle x_t, G_t \rangle\Big).$$

Hence, then using only $x_{t-1}, x_t, G_t$, ELRA computes at most three scalar products each step (the two above and $||G_t||^2 = \langle G_t, G_t \rangle$) and one vector subtraction $x_{t+1} = x_t - \alpha_t \cdot G_t$. In the retrace case even less is needed, as the norm $||G_{t-1}||$ is already computed and no scalar product

---

[8]$bs_{min}$ is a hyperparameter to be chosen for each problem.

is needed. Moreover, ELRA can be implemented such that at any time only two vectors are in the GPU-memory, as $x_{t-1}$ is only needed for the computation of $\langle x_{t-1}, G_t \rangle$ and could be stored in CPU-memory otherwise. This would require the following three additional memory transfers $x_t \to CPU$, $x_{t-1} \to GPU$, $x_t \to GPU$ (while $G_t$ stays in $GPU$[9]) which would roughly increase the overall computation time by 10%.

### 3.7 MEAN VALUE BOOSTING

We noticed that ELRA tends to oscillate round the optimal descent path (see Fig. 2b and 4). It follows that the mean $\overline{x} = \frac{1}{n}(\sum_{k=1}^{n} x_{T+k})$ of points $x_t$ for a certain number $n$ of steps (e.g. an epoch) can give better results, i.e. $f(\overline{x}) < f(x_{T+n})$. However, it is not beneficial within the optimization process to replace $x_{T+n}$ with $\overline{x}$, as $\overline{x}$ is relatively to the optimal descent path still further up than $x_{T+n}$ (in Fig. 2b, $\overline{x}$ is roughly in the middle, while $x_{T+n}$ is at the back, at the end of the green arrow). Yet in the final epochs, calculating $\overline{x}$ and $f(\overline{x})$ can boost the final result. Alternatively, using only $\overline{x}$ for the test evaluations can give better results faster. We provide for our experiments $f(\overline{x})$ for every epoch to illustrate the possible benefit.

## 4 RESULTS

As shown above (see section 2), we have a mathematical justification for our approach. Yet, giving guaranteed convergence rates for ELRA is intractable with current methods (even for convex landscapes), due to the adaptive nature of the learning rate $\alpha$. Thus we rely on experiments to show the usefulness of ELRA. However, comparison with other optimizers poses the following problem: to prevent unfair representation, the other optimizers should be run with optimal parameters. Yet finding these can be very costly. Hence we restrict ourselves to only few popular optimizers for comparison. We conducted low dimensional mathematical experiments and high-dimensional experiments with neural networks for image classification and large language models. The latter are all executed for multiple random initializations/seeds, as gradient descent methods show partially chaotic behaviour. However, for cost reasons (limitations of an academical budget) we restrict ourselves to 10 different initializations per experiment (except for Wide-ResNet, where we conducted only 6 runs, and Tiny-ImageNet/ImageNet with 3+4 or 1+2 runs). We provide graphics using the median and give the mean of the best values over all runs together with the standard deviation. We stress that no scan of the seed space was performed for ELRA, nor hyper-parameter tuning via validation data!

### 4.1 MATHEMATICAL 2D EXPERIMENTS

As proof of concept and to explore certain standard problems in gradient descent, we first show 2-dimensional results on saddle points, bowls/parabolas and the Rosenbrock function.

#### 4.1.1 SADDLE POINTS

Saddle points (where $\nabla f(x) = 0$ but $f(x)$ is not a local max/min) can pose problems in gradient descent methods, as the gradient becomes arbitrarily small near them, which might lead to catastrophic speed loss. Generically, in suitable coordinates, these saddle points look locally like $x = (0, 0)$ for $f(x) = x_1^2 - x_2^2$ (see Math. Suppl., eq. (6)). However, for a given data representation, it is more likely that the coordinates near a saddle are slightly rotated. We looked at the performance of the optimizers AdaDelta, Adam (with $\alpha = 0.01, \beta_1 = 0.9, \beta_2 = 0.999$), and our optimizer ELRA near the standard saddle $f(x) = x_1^2 - x_2^2$ starting at[10] $x_0 = (1, 10^{-9})$ and the problem rotated by $45°$ covering the two extremal situations. Fig. 3 (Left) shows the value of $f$ over steps $t$. The dashed lines belong to the rotated situation. The fastest solver is ELRA, which has the same graph with or without rotation, thereby demonstrating its inherent rotational invariance. AdaDelta and Adam are slower and suffer significantly from $45°$-rotation, as it renders the component wise modification of the Ada-family useless. Fig. 3 (Right) illustrates the paths in the $x_1$-$x_2-$plane chosen by the different optimizers. One sees that ELRA follows quickly the gradient direction, while the Ada-family either try to avoid the saddle directly (unrotated situation) or follow slowly the gradient direction. This

---

[9]The scalar product is still computed in the GPU.

[10]All true gradient descent methods fail when starting at $(1, 0)$.

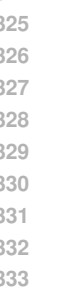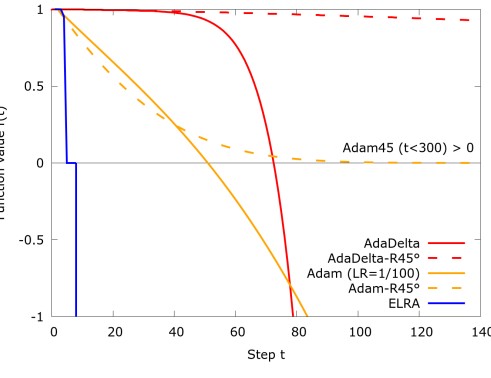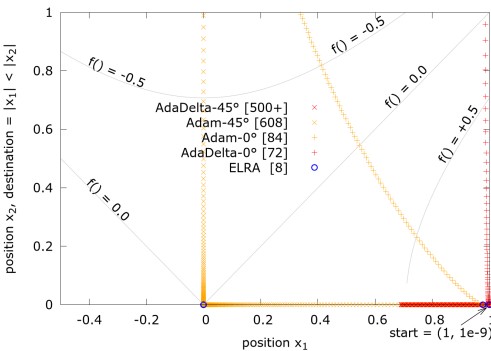

Figure 3: Behaviour of optimizers near saddle $f(x) = x_1^2 - x_2^2$ and effect of $45°$-rotation. Left: steps $t$ needed to reach $< -1$, Right: optimization paths ($\boldsymbol{+} \simeq 0°$, $\boldsymbol{\times} \simeq 45°$) in $x_1$-$x_2-$plane. Note: 4 of the 8 steps of ELRA (blue) are indistinguishable near $(0, 0)$.

shows one drawback of conditioning individual axis weights within the Ada-family. It illustrates also that the different optimizers often find different local/global minima.

#### 4.1.2 BOWLS AND ROSENBROCK

As a second class of mathematical experiments, we considered higher dimensional parabolas (so called bowls), i.e. functions of the form $f(x) = \sum_i c_i \cdot x_i^2$, and the infamous Rosenbrock function $f(x) = (1-x_1)^2 + 100(x_2 - x_1^2)^2$. Bowls provide the simplest non-trivial functions for convex optimization, while the Rosenbrock function with its curved valley is a difficult standard optimization problem. Here, we used for Adam $\alpha = 0.05, \beta_1 = 0.8, \beta_2 = 0.9$ and for RMSprop $\alpha = 0.05$.

Table 1: Steps $t$ to reach $f(x_t) < \varepsilon$ from $x_0 = (-5.75, 1.75)$ for the bowl $f(x) = 3x_1^2 + 24x_2^2$

| accuracy | Adam | RMSprop | ELRA |
|---|---|---|---|
| $\varepsilon = 10^{-1}$ | 128 | 142 | **9** |
| $\varepsilon = 10^{-6}$ | 184 | $\infty$ | **12** |

Table 2: Steps $t$ to reach $f(x_t) < 1$ from start point $x_0$ for the Rosenbrock function $f(x) = (1 - x_1)^2 + 100(x_2 - x_1^2)^2$

| start point | Adam | RMSprop | ELRA |
|---|---|---|---|
| $(-3, -2)$ | 208 | 176 | **10** |
| $(-11, 121)$ | $> 10^4$ | $> 10^4$ | **300** |

The Tables 1 and 2 give the minimal number of steps $t$ needed for the different optimizers to reach a certain threshold for $f(x_t)$. One sees that for these examples (together with the saddle from above) ELRA is by far the fastest and for Rosenbrock with bigger starting points, it is the only optimizer.

### 4.2 NEURAL NETWORKS

We conducted numerous experiments with neural networks for image classification, involving the 6 training data sets MNIST, Fashion-MNIST, CIFAR-10, CIFAR-100, Tiny ImageNet and ImageNet and 6 different neural networks ranging from tiny ($\sim$ 8k parameters) to substantial ($\sim$ 36 mil. parameters). We use batch shuffling after each epoch and only minimal data augmentation: no augmentation for MNIST, only random horizontal flips for Fashion-MNIST and random horizontal flips plus PyTorch's RandomCrop with padding=4 in reflect mode for CIFAR-10, CIFAR-100, (Tiny)ImageNet. The training data sets and neural networks used have the following specifications:

- MNIST: $(60+10)$k pictures ($28\times28$ pixels, gray-scale) of handwritten single digits, network: primitive fully connected network with 1 hidden layer (10 neurons) and ReLU-activation, runs: 10 each

- Fashion-MNIST: $(60+10)$k pictures ($28\times28$ pixels, gray-scale) of fashion items of 10 different classes, network: 3-layer convolutional network FashionCNN, runs: 10 each

- CIFAR-10: $(50+10)$k images ($32\times32$ pixels, RGB color) of objects of 10 different classes, networks: standard residual neural networks ResNet18, ResNet34 (cf. He et al. (2016)), and Wide-ResNet-28-10 (cf. Zagoruyko & Komodakis (2017)), runs: 10 or 6 (Wide-R) each

- CIFAR-100: $(50+10)$k images ($32\times32$ pixels, RGB color) of objects of 100 classes  network: ResNet18, runs: 10 each

- TinyImageNet: $(95+5)$k images ($64\times64$ pixels resized to $256\times256$ and then croppped to $224\times224$, RGB color) of objects of 200 classes, net.: ResNet18, runs: 3+4 (noWD+WD)

- ImageNet: $(1200+81)$k images ($256\times256$ pixels, RGB color) of objects of 1000 classes from the ImageNet challenge 2012 (ILSVRC2012), network: ResNet50, runs: 1+2 (noWD+WD)

We conducted all experiments with and without weight decay ($wd = 0.9997$ or $wd = 0.9999$ for (Tiny)ImageNet) and with and without the gradient decay feature (cf. section 3.4). The initial batch size was $bs_0 = 2\times32$, except for TinyImageNet ($bs_0 = 3\times32$) and ImageNet ($bs_0 = 4\times32$). The experiments lasted for 100 epochs without weight decay and for 200 epochs with weight decay, with ImageNet being again the exception - lasting only 50 epochs.

For comparison, we conducted the experiments without weight decay also for the popular optimizers Adam and Lion, with batch size $bs = 256$, default $\beta_1, \beta_2$ and constant learning rates $\alpha = 10^{-3}$ (Adam) and $\alpha = 10^{-4}$ (Lion). Our results with weight decay are compared with training results for stochastic gradient descent (SGD) taken from DeVries & Taylor (2017), who used $bs = 128$, a Nesterov-momentum with $\beta = 0.9$, $wd = 5 \cdot 10^{-4}$ and a learning rate schedule, which reduced $\alpha$ from 0.1 by a factor of 5 after 60, 120 and 160 epochs. The overall number of epochs and the data augmentation are identical to ours.

The following plot (Fig. 4) shows the typical training behavior of the median test accuracy in the experiments, illustrated by the CIFAR-10 experiments on ResNet18. Note that each ELRA-experiment

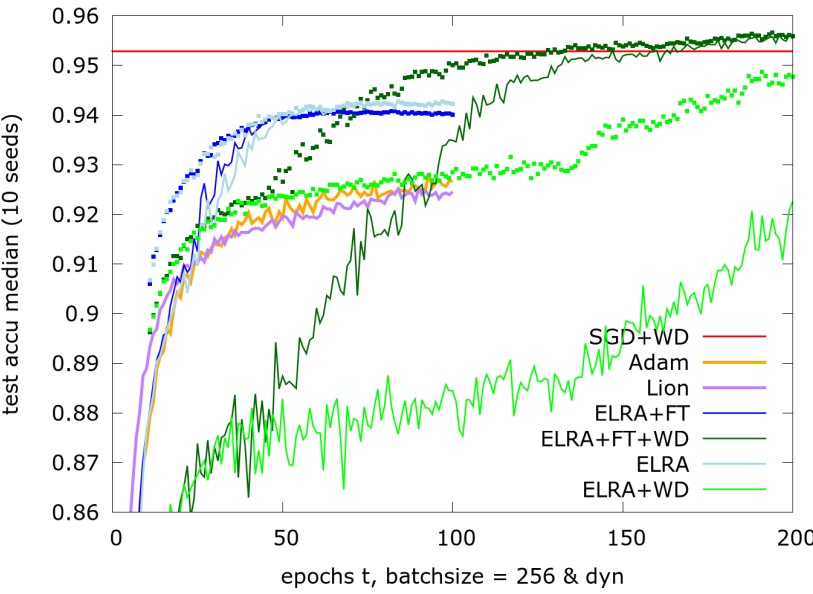

Figure 4: CIFAR-10, ResNet18: Median Test-accuracy for 100 epochs (without wd) and 200 epochs (with wd). The bolds lines for ELRA are with Boost (cf. Sec. 3.7), the thin lines without.

has two lines, one with Boost (cf. Sec. 3.7), one without. With the exception of ELRA+WD (weight decay without gradient decay), the boosted and unboosted values meet at the end of the training (for ELRA+WD they meet around the epoch 300). Hence, we give in the following only the boosted values, as it makes no difference. The clearly visible bends in the curves for ELRA are the points where for the first time the batch size increment is triggered by the algorithm.

The tables 3-5 show the mean over the best test accuracies with standard deviation for each experiment. The data for comparision in tables 4 and 5 are taken from the literature. Note that ELRA greatly profits from the use of weight decay, but keep in mind that the training time is also doubled. Moreover, the gradient decay feature is only really beneficial with weight decay and only for networks with strong overfitting (Fashion-MNIST, CIFAR, cf. section D).

Table 3: Best test accuracy (%) without weight decay

| network | ELRA | ELRA+FT | Adam | Lion |
|---------|------|---------|------|------|
| MNIST | 94.25±0.21 | 94.41±0.10 | 93.90±0.56 | 93.82±0.49 |
| Fashion-MNIST | 92.58±0.13 | 92.71±0.11 | 92.37±0.14 | 92.48±0.13 |
| C10+R18 | 94.34±0.14 | 94.11±0.07 | 93.01±0.10 | 92.77±0.10 |
| C10+R34 | 94.54±0.19 | 94.09±0.42 | 93.26±0.07 | 93.04±0.13 |
| C10+WRN-28-10 | 95.11±0.11 | 94.86±0.15 | - | - |
| C100+R18 | 73.99±0.38 | 74.05±0.30 | 70.22±0.31 | 70.24±0.16 |

Table 4: Best test accuracy (%) with weight decay

| network | ELRA | ELRA+FT | SGD[11] |
|---------|------|---------|---------|
| MNIST | 94.70±0.25 | 94.67±0.19 | - |
| Fashion-MNIST | 92.77±0.08 | 93.31±0.10 | - |
| C10+R18 | 94.83±0.65 | 95.77±0.16 | 95.28±0.21 |
| C10+R34 | 94.77±0.77 | 95.80±0.08 | - |
| C10+WRN-28-10 | 96.21±0.10 | 96.21±0.03 | 96.13±0.08 |
| C100+R18 | 78.12±0.16 | 79.35±0.16 | 77,54±0.31 |

Table 5: Best test accuracy (%) for (Tiny)ImageNet

| network | ELRA | ELRA+FT | IRRCNN[12] | unknown[13] |
|---------|------|---------|------------|-------------|
| TIN-200 | 56.10±0.61 | - | 52.23 | - |
| TIN-200+WD | 59.47±0.59 | - | 52.23 | - |
| ImgNet-1k | 70.69±0 | 70.94±0 | - | 76.00 ±0.0 |
| ImgNet-1k+WD | 75.54±0.01 | 73.64±0.22 | - | 76.00 ±0.0 |

## 4.3 ANALYSIS

For the almost trivial problems MNIST and Fashion-MNIST, the advantage of the ELRA optimizer over whose from the Ada-family is (at least without weight decay) only marginal. Yet the lead is overwhelming for CIFAR. On the other hand, ELRA shows similar (albeit slightly better) results to SGD (with learning rate scheduler), when weight decay is used. This suggests that as an optimizer, ELRA could be considered a variant of SGD, yet with adaptive learning rate $\alpha$ and batch size, which eliminates the hand-tuning of a learning rate scheduler. For SGD it has been observed that, albeit being slower, it tends to yield minimizers, which generalize better to the test data. This is explained by SGD having noisier optimization paths, which helps to escape steep local minima, which generalize less optimal. See Zhou et al. (2020) and Huang et al. (2019) for some explanations of this effect. Our experiments suggest that ELRA shares this behaviour with SDG.

We note that the gap between ELRA and Adam and Lion can be reduced with the use of aditional features such as warm-up, learning rate scheduler and heavy data augmentation. However, all these need additional calibration runs or an experienced programmer. Strongly adaptive learning rates and

---

[11] Results taken from DeVries & Taylor (2017)

[12] Results taken from vpn.th-wildau.deAlom2020, trained for 70 epochs

[13] Results taken from Dauphin & Cubuk (2021), trained for 90 epochs

batch sizes seem to have the potential to eliminate the need for these extra features.

Finally, the experiments with (Tiny)ImageNet show that ELRA also works with much larger networks/training data sets, yielding comparable results to the literature (without the use of more involved modern training schemes/data augmentations).

### 4.4 LARGE LANGUAGE MODELS – BABYLLAMA

Recently, we also started testing ELRA on Large Languages models, using the BabyLlama model with $\sim$15 mil. parameters. The problem here is, that ELRA needs at the moment a very small initial batch size $bs_0 = 1 \times 16$. This makes performance comparison with popular optimizers difficult, as they use typically a much larger batch size (e.g. 512 or 1024). On the one hand, runs with smaller batch sizes fit more easily into the GPU, on the other hand larger batch sizes can make better use of parallel computations. Here, we need to invest more time into the development of an implementation which combines the adaptive batch size with the use of multiple GPUs. However, our initial experiments suggest that ELRA yields similar test losses as AdamW (with $\alpha = 5 \cdot 10^{-4}$, $wd = 10^{-1}$, $\beta_1 = 0.9$, $\beta_2 = 0.95$, $bs = 512$): After 100.000 steps (corresponding to $\sim$51 mil. tokens), AdamW reached a validation loss of 1.08, while ELRA (with $bs_0 = 16$, no $wd$) reached after 800.000 steps (corresponding to $\sim$12.8 mil. tokens, or 25% of AdamW's run) a validation loss of 1.156 (after the same amount of tokens, AdamW had a validation loss of 1.187).

## 5 LIMITATIONS

Our implementation still leaves much room for improvements. For instance our code and consequently the computational resources needed would benefit from ELRA specific adaptations of the PyTorch or Jax architecture, such as providing by default the function value $f(x_t)$ to the optimizer and the implementation of a function that computes simultaneously the scalar products $\langle G_t, G_t \rangle$, $\langle G_{t-1}, G_t \rangle$. Moreover, a flexible data loader for varying batch sizes together with multi-GPU computation could give ELRA a significant speed boost.

The last part is particular important, as ELRA tends to work better with smaller than usual initial batch sizes $bs_0$. As we increase $bs$ at the moment solely by accumulation, we cannot use the speed gain coming from computing larger batches on multiple GPUs.

Finally, ELRA has some relevant hyperparameters to be chosen for each experiment: The initial batch size, the length of the batch size control cycles, (if needed) a fixed weight decay and (again if needed) the warm-up condition $\alpha_t \cdot ||G_t|| \geq c$. Also, the use of the gradient decay feature is optional. Here, some future guidelines for choices should be developed.

## 6 CONCLUSION

We presented the novel, simple, self-adjusting, robust and fast optimizer ELRA with linear dimensional scaling, rotational invariance and without momentum. Typical runs on mathematical standard problems and statistical tests on neural networks for the (Fashion)MNIST, CIFAR and (Tiny)ImageNet data sets with several initializations showed better final test accuracies then the popular optimizers Adam, Lion or SGD with hand-tuned optimal parameters! Moreover, the adaptive learning rates and batch sizes seem to eliminate the need for hand-tuned learning rate schedulers and (to some extent) heavy data augmentation, thus leading to greater universality and possible out-of-the-box usage.

We believe that nobody has thought about trying steep and fast $\alpha$-adaptions before due to the following reasons: for small dimensions good solvers exist (often using matrix inversions, e.g. the Levenberg–Marquardt algorithm), mathematical optimizers strive for provability (which restricted until recently to constant $\alpha$: compare Nesterov (2018) and Grimmer (2023)) and previous conditions (Armijo) for updating $\alpha$ are too expensive in high dimensions.

Finally, better control systems for the learning rate, batch sizes and soft restarts promise to further increase performance and universality (see Future works below).

We strongly believe that the above ideas will create a completely new research field in gradient descent-based optimization.

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

# A METHODS DETAILS

## A.1 SUMMARY

Table 6: Properties of Adaptive Optimizer Features

| Main feature | Level of maturity | General applicable |
|---|---|---|
| dynamic alpha | great (fast, robust, universal) | yes, simple |
| dynamic batch-size | good (fast, robust, universal) | yes, independent of any SGD optimizer |
| (final) boosting | great, improves most results | could help Adam (etc.) in final epochs |
| soft restart | good, reverts failed step | simple, more robustness with high $\alpha$ |

## A.2 ESTIMATING $\alpha$ USING A PARABOLA ANSATZ

To get the update formula for $\alpha$ of ELRA, we consider $f$ only along the straight line through $x_{t-1}$ and $x_t$, whose direction is $x_t - x_{t-1} = -\alpha_{t-1} G_{t-1}$. We assume that $f$ along this line is a parabola, i.e. $f(x) = ax^2 + b$, where we chose (in practice unknown) coordinates such that $x = 0$ is the minimizer of $f$. In this setting, the derivatives of $f$ are:

$$2ax_{t-1} = f'(x_{t-1}) = \partial_{G_{t-1}} f(x_{t-1}) = ||G_{t-1}||, \quad 2ax_t = f'(x_t) = \partial_{G_{t-1}} f(x_t) = \frac{\langle G_{t-1}, G_t \rangle}{||G_{t-1}||},$$

where $\partial_{G_{t-1}}$ is the directional derivative of $f$ with respect to $G_{t-1}$. Together with $x_t - x_{t-1} = -\alpha_{t-1} G_{t-1}$, we get

$$2a(x_t - x_{t-1}) = \frac{\langle G_{t-1}, G_t \rangle}{||G_{t-1}||} - ||G_{t-1}|| = \frac{\langle G_{t-1}, G_t \rangle - ||G_{t-1}||^2}{||G_{t-1}||} \quad \Rightarrow \quad a = \frac{\langle G_{t-1}, G_t \rangle - ||G_{t-1}||^2}{-2\alpha_{t-1} \cdot ||G_{t-1}||^2}$$

As we want $x_t - \alpha_t \cdot f'(x_t) = x_{t+1} = 0$ to be the minimizer of $f$, we obtain with $x_t = \frac{f'(x_t)}{2a}$ that

$$\alpha_t = \frac{x_t}{f'(x_t)} = \frac{1}{2a} = \alpha_{t-1} \cdot \frac{||G_{t-1}||^2}{||G_{t-1}||^2 - \langle G_{t-1}, G_t \rangle} = \alpha_{t-1} \cdot \left( 1 + \frac{\langle G_{t-1}, G_t \rangle}{||G_{t-1}||^2 - \langle G_{t-1}, G_t \rangle} \right)$$

$$= \alpha_{t-1} \cdot \left( 1 + \frac{\cos_t}{||G_{t-1}||/||G_t|| - \cos_t} \right), \tag{5}$$

where we used $\langle G_{t-1}, G_t \rangle = \cos_t \cdot ||G_{t-1}|| \cdot ||G_t||$ in the final step.

# B FUTURE WORK

Opening a new field creates lots of opportunities for continuation. Here we mention some of the most promising directions:

- $\alpha = \alpha \cdot (1 + \cos \cdot g(x))$ is the general update scheme for $\alpha$ obtained from our idea of orthogonal gradients equation 3. Here, $g$ can be any function with $g(x) > 0$. What is the best $g$? Different answers for different problems?

- Problem specific fine tuning (selected hyper-parameters) is possible and could give further improvement:

    - fixed bounds for $\alpha$ (i.e. $10^{-7} < \alpha < 10^{-1}$) based on statistics gathered during current run. Could speed up ELRA (fewer soft restarts, shorter time to recover from restart)

- Further applications: electronic structure optimizations, protein folding, molecular dynamics, finite element methods

- Possible landscape characterization as a side-result

## C MATHEMATICAL SUPPLEMENTS

### C.1 EXTREMAL POINTS SIT INSIDE QUADRATIC SURROUNDING

In principle, critical points $x_0$, such as local/global minima and saddle points, can be degenerate, i.e. the Hessian at $x_0$ can have 0 as an eigenvalue. However, functions with all critical points non-degenerate, so called Morse functions, are the generic situation, meaning that they form an open and dense subset within $C^2(\mathbb{R}^n)$, see Milnor (1965). So figuratively speaking, "almost all" two times continuously differentiable functions have only non-degenerate critical points. For these functions $f$, we find then by Taylor expansion, that they are locally dominated by their Hessian, i.e. they behave locally around critical points like quadratic functions:

$$f(x) = \sum_{i=1}^{n} c_i \cdot x_i^2, \qquad c_i \in \{+1, -1\}. \tag{6}$$

### C.2 RANDOM NOISE CONVOLUTION REMOVES DISCONTINUITIES IN GRADIENTS

In some applications, the function $f$, which we want to optimize, is not differentiable, such as $f(x) = ||x||$ or $f(x) = \max\{x, 0\}$. Then, the gradient is not everywhere defined and most optimization methods suffer. However, if the data contains some random noise, i.e. the function $f$ is slightly blurred, then we can expect differentiability. Indeed, the effect of random noise can be thought of as convoluting $f$ with a probability density function $\phi$, such as the density of the normal distribution $\phi(x) = exp(-x^2/2\sigma^2)/(\sigma\sqrt{\pi})$ (if the blurring can be arbitrarily large) or a density with finite support, if the blurring is limited. Now, if $\phi$ is continuously differentiable and $f$ integrable or locally integrable (for finite support), then it is a well known fact that the convolution $f * \phi$ is also differentiable with differential

$$\partial_{x_i}(f * \phi)(x) = \partial_{x_i} \int_{\mathbb{R}^n} f(t) \cdot \phi(x-t)\mathrm{d}t = \int_{\mathbb{R}^n} f(t) \cdot \partial_{x_i}\phi(x-t)\mathrm{d}t = (f * \partial_{x_i}\phi)(x). \tag{7}$$

Especially DNN learning should be affected by noise from the input and from batching, resulting in smooth landscapes.

## D ADDITIONAL PERFORMANCE PLOTS

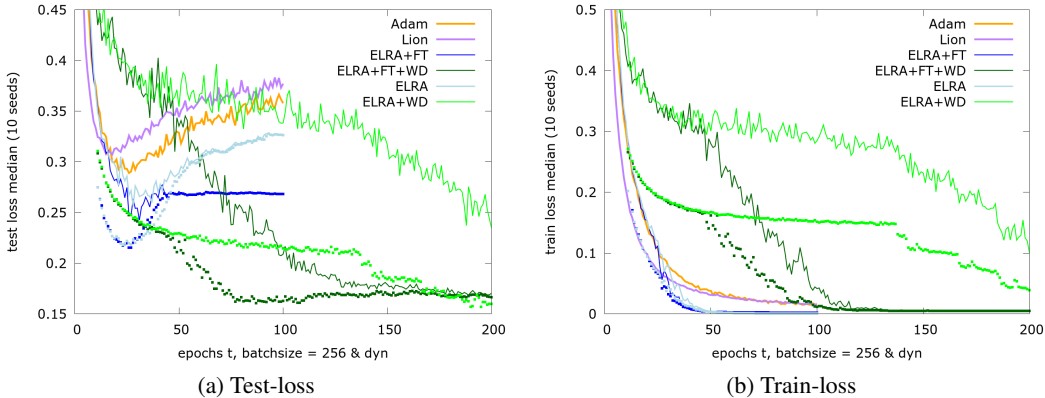

(a) Test-loss         (b) Train-loss

Figure 5: Median Test-/Train-loss over 100/200 epochs for CIFAR-18.

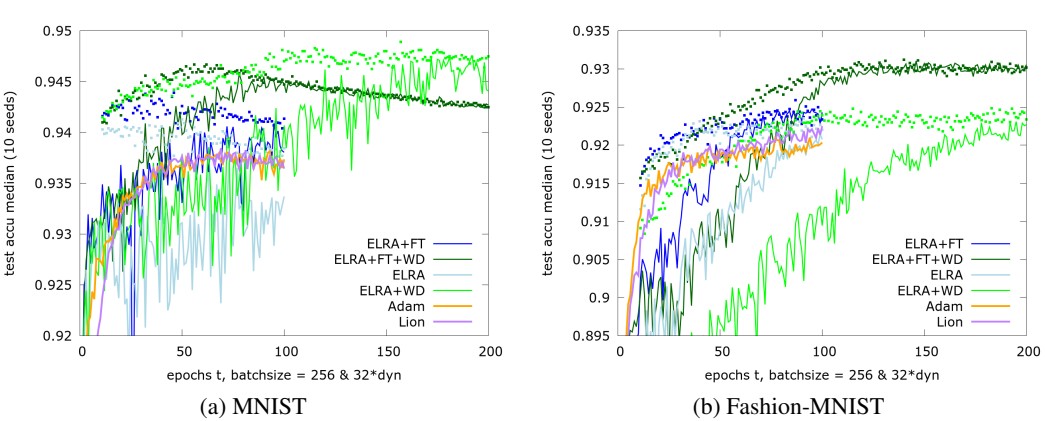

(a) MNIST

(b) Fashion-MNIST

Figure 6: Median Test-accuracy over 100/200 epochs for (Fashion-)MNIST.

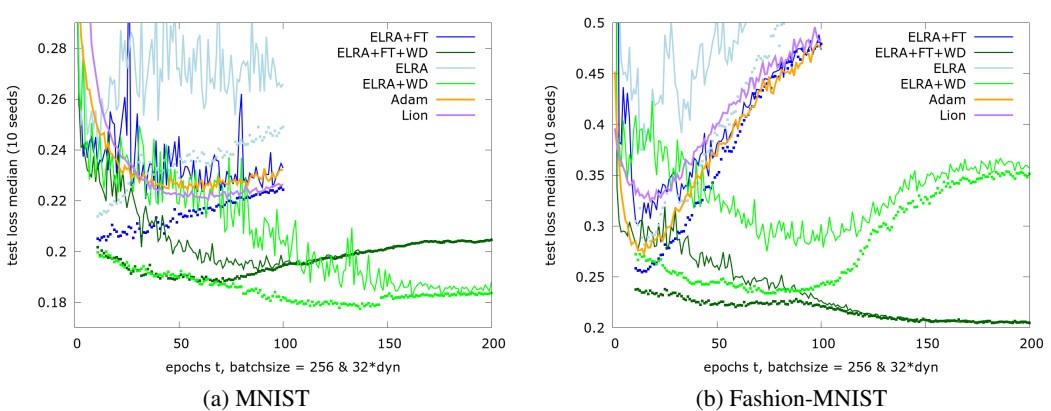

(a) MNIST

(b) Fashion-MNIST

Figure 7: Median Test-loss over 100/200 epochs for (Fashion-)MNIST.

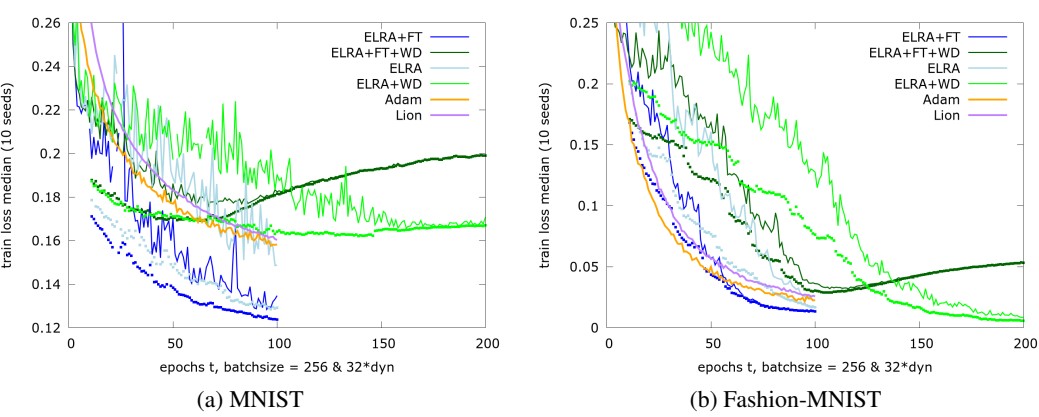

(a) MNIST

(b) Fashion-MNIST

Figure 8: Median Train-loss over 100/200 epochs for (Fashion-)MNIST.

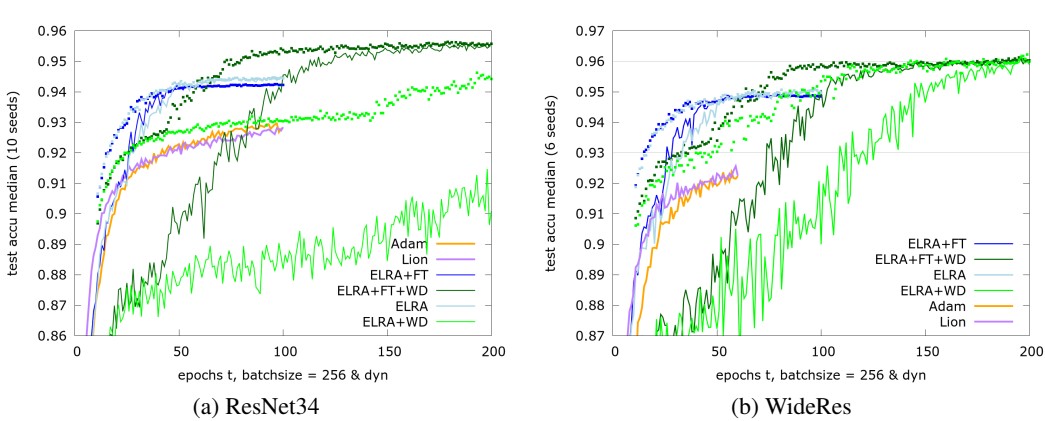

(a) ResNet34

(b) WideRes

Figure 9: Median Test-accuracy over 100/200 eps. for CIFAR-10 on ResNet-34/Wide-ResNet-28-10.

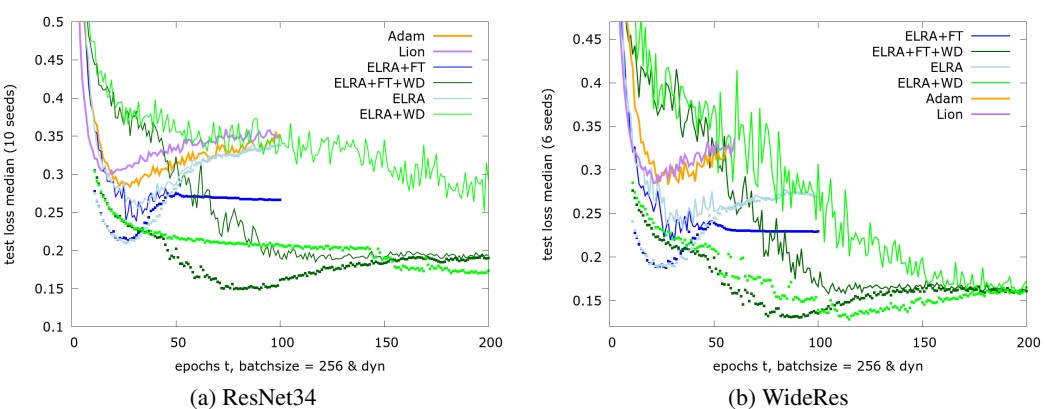

(a) ResNet34

(b) WideRes

Figure 10: Median Test-loss over 100/200 eps. for CIFAR-10 on ResNet-34/Wide-ResNet-28-10.

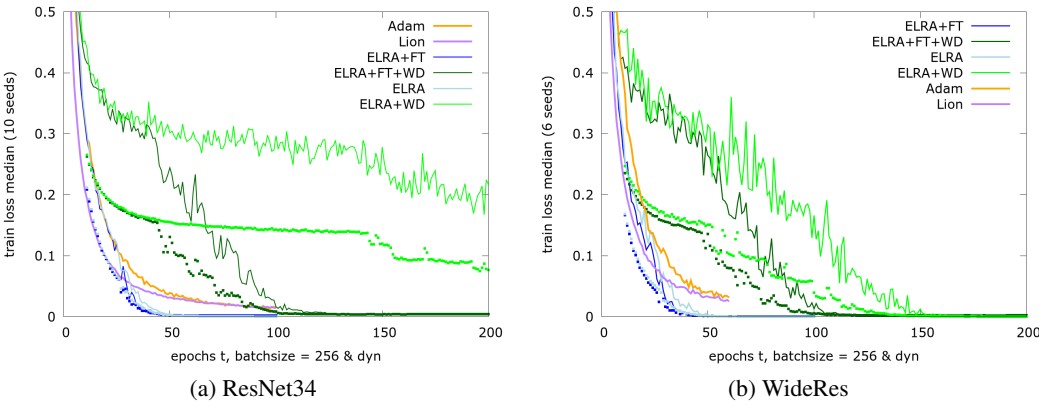

(a) ResNet34

(b) WideRes

Figure 11: Median Train-loss over 100/200 eps. for CIFAR-10 on ResNet-34/Wide-ResNet-28-10.

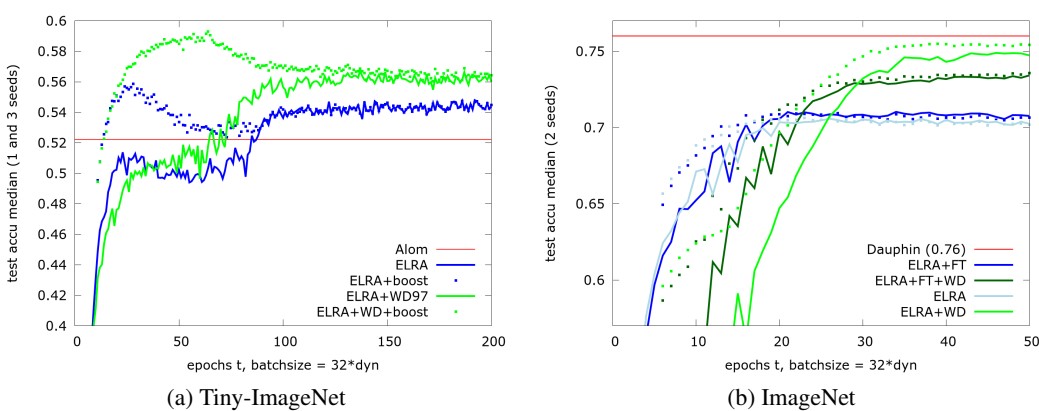

(a) Tiny-ImageNet

(b) ImageNet

Figure 12: Median Test-accuracy over 200/50 epochs for TinyImageNet/ImageNet using ResNet18/ResNet50, including reference values.

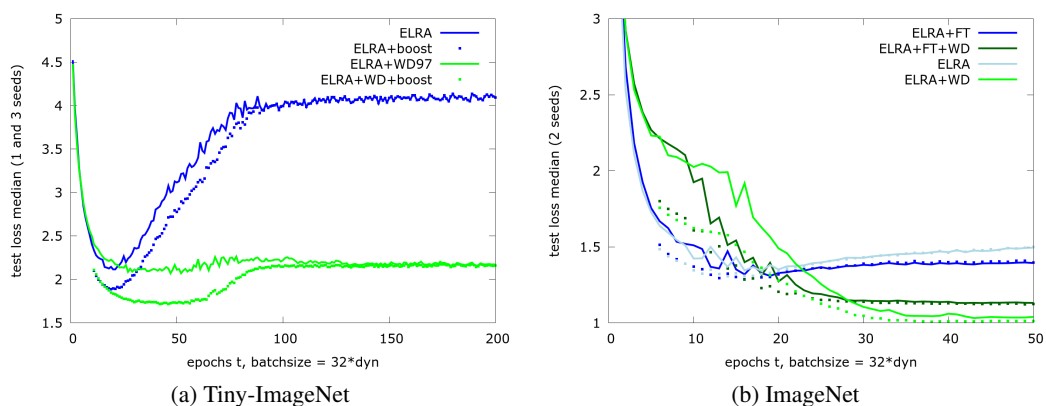

(a) Tiny-ImageNet

(b) ImageNet

Figure 13: Median Test-loss over 200/50 epochs for TinyImageNet/ImageNet using ResNet18/ResNet50.

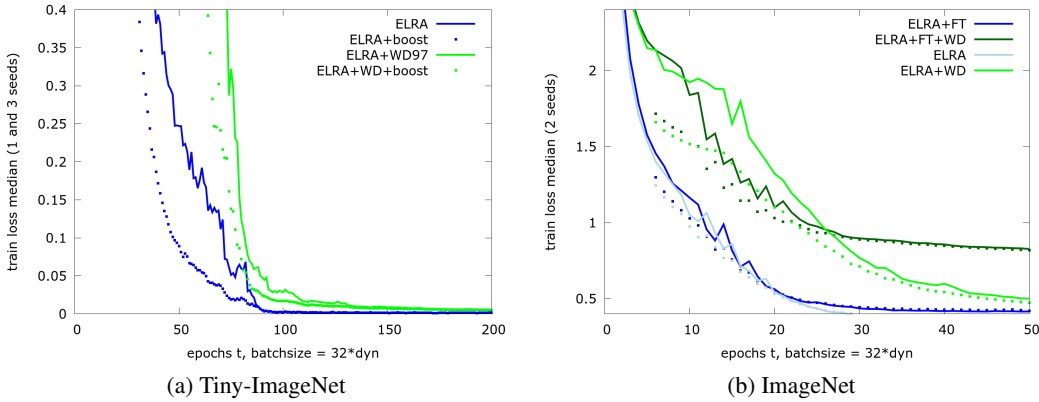

(a) Tiny-ImageNet

(b) ImageNet

Figure 14: Median Test-loss over 200/50 epochs for TinyImageNet/ImageNet using ResNet18/ResNet50.

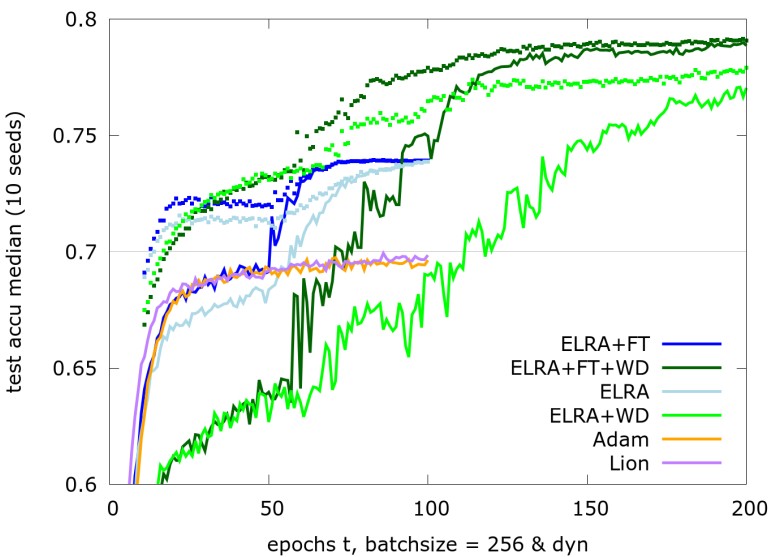

Figure 15: Median Test-accuracy over 100/200 epochs for CIFAR-100 on ResNet-18

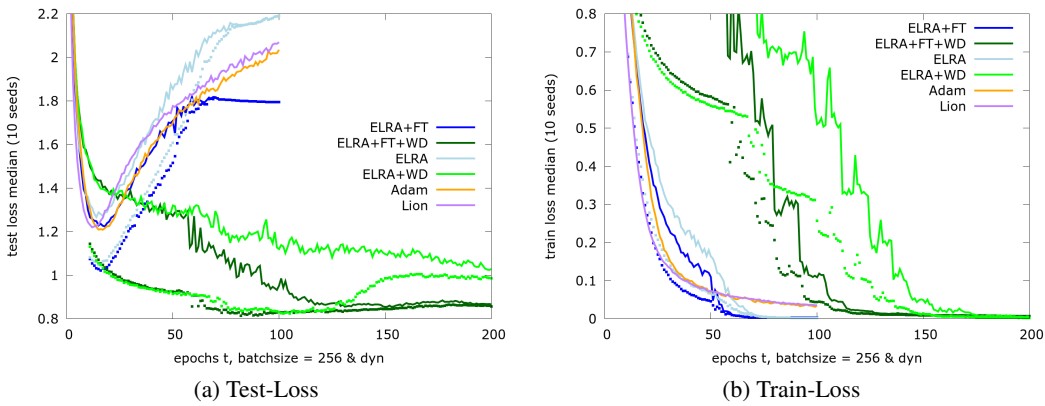

(a) Test-Loss

(b) Train-Loss

Figure 16: Median Test/Train-loss over 100/200 epochs for CIFAR-100 on ResNet-18

