# OpenReview forum: "A Gradient Descent Optimizer with auto-controlled large Learning Rates, dynamic Batch Sizes and without Momentum"
_ICLR.cc/2025/Conference — Submitted to ICLR 2025_

### Official Review · Reviewer_BZbF · 2024-10-30

**Soundness:** 1
**Presentation:** 1
**Contribution:** 2
**Rating:** 3
**Confidence:** 3

**Summary:**

This work presents a new fast gradient based momentum-free optimizer algorithm with dynamic learning rate and dynamic batch size. They evaluated their algorithm on several benchmarks and made some basic empirical achievement.

**Strengths:**

* The authors have fused several accelerating tricks in the field of optimization into their optimizer, it seems to surprisingly work well.
* The authors provide interesting derivations for their design of the optimizer

**Weaknesses:**

* It needs to clarify that, since this optimizer introduces adaptive learning rate for each parameter, then momentum-free shouldn’t be an advantage of this work at efficiency, the cache of learning rates is equivalent to cache momentum practically.
* In Section 2, the definition of the local minimum of $\alpha$ is unclear, as the authors assume that $f$ is convex, whereas it does not straightly lead that $f(x_t)$ is convex for $\alpha$, I am not sure about the effectiveness of devoting “local minimum” for faster optimization theoretically.
* The key derivation in Section 2 first appeared in the delta-bar-delta method in [1], in which it starts a series of following works, please at least review these works in the paper, and compare how the key idea of ELRA demonstrates advancement. In my opinion, the reasons why the old trick doesn’t last long in the application of optimization may be various. However, this work lacks a considerable review to the prior works.
* The efficiency claim in line 257 to line 258 sounds unprofessional: This gives ELRA a roughly 10% computation advantage over Adam and Lion, when used with the same batch size. The authors should at least provide some experimental analysis to prove it.
* It’s not clear what problem or challenges the study mainly aims to address, I checked 7 subsections in Section 3, and found no necessary or professional reason for introducing such or that trick in this optimizer. It seems like an addition of several existing works, but with poor writing.
* Showcasing “fast” needs comprehensive experiments, evaluations on some toy datasets seem far away from the word “enough”. Could the authors add some speed analysis on those real-world benchmarks compared with baseline optimizers like Adam, Lion, SGDm?
* Typo: line 479: Our experiments suggest that ELRA shares this behaviour with SDG.
* I didn’t find out the definition of “ELRA+FT”, please define it somewhere conspicuous, since it looks like this line performs best in the results but I could not find what it is.
* In conclusion, this paper has a limitation in its presentation, some words seem unprofessional. I suggest the authors to re-organize the whole writing to tell a better story and show a better result.

References:
[1]: Bernard Widrow, Marcian E Hoff, et al. Adaptive switching circuits. In IRE WESCON convention record, volume 4, pages 96–104. New York, 1960.

**Questions:**

Please answer the issues and questions in the Weakness and point out my potential misunderstandings. I am happy to discuss and enhance my rate.

---

> ### Author Response · Authors · 2024-11-21
> **Clarification of fundamental misunderstanding**
>
> Dear reviewer,
> unfortunately, there has been a big misunderstanding of how our idea/algorithm works: we do not propose to have individual learning rates $\alpha$ for each parameter but one learning rate $\alpha$ (a single number) for the whole gradient. In so far we neither have the same cache use as with momentum nor is our algorithm a direct derivative of the delta-bar-delta method. We are very sorry if this is not clearly enough presented in the article and hope that you reread  our contribution under this new light.
> Moreover, the aim of our method is to provide an optimizer which has to be fine-tuned less (ideally not at all) while performing similar to fined-tuned optimizers. This is the reason for the adaptive batch size, adaptive learning rate and no momentum.
> thirdly, I could not find the delta-bar-delta method in the cited article by Bernard Widrow at al. Could it be that you meant the article "Increased rates of convergence through learning rate adaptation" 1988 by Robert A Jacobs?
>
> We will address your other remarks and questions tomorrow or over the weekend, but we felt that we should clear this fundamental misunderstanding first.

---

> ### Author Response · Authors · 2024-11-22
>
> >In Section 2, the definition of the local minimum of $\alpha$ is unclear, as the authors assume that $f$ is convex, whereas it does not straightly lead that $f(x_t)$ is convex for $\alpha$, I am not sure about the effectiveness of devoting “local minimum” for faster optimization theoretically.
>
> We do not assume in Section 2 that $f$ is convex. We use in Section 3 as an ansatz for the $\alpha$-update that $f$ is a parabola to fix an explicit update formula, meaning that if the function were a parabola, then our $\alpha$-update would be optimal. For neural networks this almost never holds. However, by second order Taylor expansion, one can assume locally that the loss function is approximately quadratic.
> The possible effectiveness of a local minimum goes as follows: We try to estimate in each step the best (i.e. local minimizer) learning rate $\alpha$ such that $f(x_{t-1}{-}\alpha G_t)$ gives (locally) the smallest possible value.
>
>
> >The key derivation in Section 2 first appeared in the delta-bar-delta method in [1], in which it starts a series of following works, please at least review these works in the paper, and compare how the key idea of ELRA demonstrates advancement. In my opinion, the reasons why the old trick doesn’t last long in the application of optimization may be various. However, this work lacks a considerable review to the prior works.
>
> We use the derivation in Section 2 only as a general guideline. Our actual new idea is the update formula for $\alpha$ presented in Section 3.1. As such, we do not want to give the impression that Section 2 is original or new. We were not aware of the relative closeness of our idea to that of the delta-bar-delta method (albeit, we do not apply it for each component individually nor do we have a fixed update rate for $\alpha$) and are grateful for pointing that connection out to us.
>
>
> >The efficiency claim in line 257 to line 258 sounds unprofessional: This gives ELRA a roughly 10% computation advantage over Adam and Lion, when used with the same batch size. The authors should at least provide some experimental analysis to prove it.
>
> You are completely right, we will eliminate this line. We remeasured the actual speed and found the following results: For CIFAR-10 with ResNet18 and batch size 128 Adam needed 12.7 sec/epoch, while ELRA needed 12.5 sec/epoch, while for batch size 32 the results are 15.5 sec/epoch compared to 14.6 sec/epoch. Moreover, the explicit time needed and the time difference is environment and problem specific.
>
>
> >It’s not clear what problem or challenges the study mainly aims to address, I checked 7 subsections in Section 3, and found no necessary or professional reason for introducing such or that trick in this optimizer. It seems like an addition of several existing works, but with poor writing.
>
> See first response.
>
>
> >Typo: line 479: Our experiments suggest that ELRA shares this behaviour with SGD.
>
> Thank you for finding that.
>
>
> >I didn’t find out the definition of “ELRA+FT”, please define it somewhere conspicuous, since it looks like this line performs best in the results but I could not find what it is.
>
> By ELRA+FT we mean with the gradient decay feature (see line 386/387), as described in Section 3.4.

---

> > ### Comment · Reviewer_BZbF · 2024-11-25
> >
> > Thank you for your rebuttals.
> >
> > As I mentioned in my reviews, this draft demonstrates a very poor presentation. Besides, the authors' rebuttals didn't address my concerns, so far during the revision, it seems like they didn't try to modify their draft for further improvements. I recommend they reconstruct this paper's whole writing to enhance the story they want to tell the AI community and supplement comprehensive experiments to emphasize the practical contribution of this work. Hence, by now, I will keep my rating score.

---

### Official Review · Reviewer_ZPXz · 2024-11-02

**Soundness:** 2
**Presentation:** 3
**Contribution:** 2
**Rating:** 3
**Confidence:** 4

**Summary:**

This paper proposes an optimization algorithm that dynamically tunes the learning rate and dynamics, based on its last two gradients. The algorithm is evaluated on neural networks with several standard benchmark datasets, and compared with SGD and Adam.

**Strengths:**

The proposed algorithm introduces not much extra computation cost while tuning the learning rate, as the tuning mechanism only depends on the norms of, and inner products between, past two gradient vectors.

There is less need in tuning the initial learning rate, compared with pure SGD. The algorithm is invariant under coordinate rotation, which could be an advantage over Adam and other Ada-family algorithms.

**Weaknesses:**

1: The design of the proposed algorithm is based on strong assumptions/conjectures which may not be true in practice. Specifically, it assumes that the loss function is a parabola in the line that goes through any two consecutive iterates. Moreover, it requires that the minimizer of the parabola happens at $x=0$. These conditions usually do not hold in modern deep learning where loss function is considered as highly non-convex, and is far from quadratic.

In addition, the paper did not verify these assumptions/conjectures in experiments.

2: The proposed algorithm does not seem to improve empirical performance (according to the experiments of the paper). Without those additional techniques (FT, WD), ELRA actually performs quite worse than SGD. The successful run includes too many other techniques (boosting, FT, weight decay, gradient decay etc), it is not clear whether it is ELRA or those accessories that lead to a relatively good performance.

3:  [about the empirical term $\kappa$]. The algorithm introduced the empirical term $\kappa$.
There is no theoretical justification of introducing it
There is no explanation of the formula of $\kappa$ (Line 151). Is it an empirical choice?
There is no (theoretical) justification of the claim “($\kappa$) neutralizes random noise effects in neural networks”. It is hard to believe a single scalar can neutralize random noise. There must be some theory to support the claim.

4: In line 123, the paper says “we expect the optimal $\alpha_t$ for $x_t$ does not vary too much from the optimal $\alpha_{t-1}$ for $x_{t-1}$. I could not see why this should be true. The algorithm makes discrete steps (some steps may be quite big), hence $x_t$ is not necessarily close to $x_{t-1}$. At least, the paper needs to experimentally verify the relation between $\alpha_t$ and $\alpha_{t-1}$.

5: As for saddle points, the paper only looked at a special type $f(x)=x_1^2-x_2^2$. However, geometry near saddle points can be much more complicated than this special case, and the analysis of the special case may not generalize.

**Questions:**

no further questions, see comments above

---

> ### Author Response · Authors · 2024-11-22
>
> >The design of the proposed algorithm is based on strong assumptions/conjectures which may not be true in practice. Specifically, it assumes that the loss function is a parabola in the line that goes through any two consecutive iterates. Moreover, it requires that the minimizer of the parabola happens at 0. These conditions usually do not hold in modern deep learning where loss function is considered as highly non-convex, and is far from quadratic. In addition, the paper did not verify these assumptions/conjectures in experiments.
>
> We do not assume that the loss function is a parabola. This is in section 3.1 just used as an ansatz to fix an explicit update formula for $\alpha$, meaning that if the function were a parabola, then our $\alpha$-update would be optimal. In the general situation this will never hold. However, by second order Taylor expansion, one can assume locally that the loss function is approximately quadratic (or linear), meaning that our ansatz should provide at least locally a good update for $\alpha$.
> Moreover, we definitely do not assume that the minimizer is at zero, we only state that one can find (theoretically), e.g. by a linear shift, coordinates such that in these coordinates the minimizer is at zero. This is just made to make the derivation of the $\alpha$-update easier and poses no problems, as the final formula is again invariant under the choice of coordinates.
>
>
> >The proposed algorithm does not seem to improve empirical performance (according to the experiments of the paper). Without those additional techniques (FT, WD), ELRA actually performs quite worse than SGD. The successful run includes too many other techniques (boosting, FT, weight decay, gradient decay etc), it is not clear whether it is ELRA or those accessories that lead to a relatively good performance.
>
> We want to note that the SGD runs use also weight decay (WD) and a learning rate scheduler, which are the best tuned values for SGD as found by the community. Moreover, boosting only increases the speed, not the final performance (as can be seen by the training-loss graphics in the appendix). Also, we do not intend to say that our learning rate update alone produces the best results (which is also true for all modern optimizers which use weight decay, 1st and second order momentum and a learning rate scheduler). Rather, we aim the provide an optimizer, where the hyperparameters are more indirect in the hope that they do not need fine tuning for every individual network. All our runs use identical hyperparameter values, except for initial batch size, initial learning rate $\alpha_0$ (which we show has no great influence, see Fig. 2) and the weight decay rate.
>
>
> >The algorithm introduced the empirical term $\kappa$. There is no theoretical justification of introducing it. There is no explanation of the formula of (Line 151). Is it an empirical choice? There is no (theoretical) justification of the claim “$\kappa$ neutralizes random noise effects in neural networks”. It is hard to believe a single scalar can neutralize random noise. There must be some theory to support the claim.
>
> There is no explicit theoretical justification for $\kappa$, as we do not have a satisfactory one at the moment. However, here are some of our ideas behind $\kappa$: Firstly, why should a single number work at all? Since our $\alpha$-update uses scalar products and increases/decreases $\alpha$ if the product is positive/negative, we are mostly concerned with noise affecting this sign, which is a 1-dimensional parameter, hence one can also expect that a 1-dim term can reduce the noise effect. Secondly, we found empirically that $\kappa\sim 1.15$ works quite good, unless $\alpha$ is already quite large ($\alpha>1$). The explicit formula for $\kappa$ is close to 1.15 for $\alpha\ll 1$ and reduces $\kappa$ to 1 for large $\alpha$.
>
>
> >In line 123, the paper says “we expect the optimal $\alpha_t$ for $x_t$ does not vary too much from the optimal $\alpha_{t-1}$ for $x_{t-1}$. I could not see why this should be true. The algorithm makes discrete steps (some steps may be quite big), hence $x_t$ is not necessarily close to $x_{t-1}$. At least, the paper needs to experimentally verify the relation between $\alpha_t$ and $\alpha_{t-1}$.
>
> Such an assumption is made by all optimizers (though not always stated explicitly), as all optimizers make discrete steps and try to predict the future from past knowledge. In non-convex landscapes (such as for modern large neural networks) it can therefore always happen that these predictions fail. Note that the convergence proofs for almost all gradient descent optimizers assume at least that $f$ is convex, while this almost never holds for neural networks. That our predictions for $\alpha_t$ are practicable is shown by the performance of ELRA.

---

> > ### Author Response · Authors · 2024-11-22
> > **Reply continued**
> >
> > >As for saddle points, the paper only looked at a special type . However, geometry near saddle points can be much more complicated than this special case, and the analysis of the special case may not generalize.
> >
> > The geometry of saddle points is, up to a change of coordinates exactly like $f(x)=\sum_{i=1}^k x_i^2 -\sum_{k+1}^n x_i^2$, i.e. like the standard saddle point we considered (see our appendix C.1 and the literature mentioned therein). Our point in 4.1.1 however is that Adam and all related optimizers are coordinate the depend with regards to saddle points, while ELRA does not depend on the specific coordinates.

---

### Official Review · Reviewer_eMGk · 2024-11-05

**Soundness:** 3
**Presentation:** 3
**Contribution:** 3
**Rating:** 5
**Confidence:** 4

**Summary:**

This paper derives a new step size schedule based on a quadratic model. The key observation is that
the optimal step size happens when current and previous gradients are orthogonal. Thus, their inner products play an important role in controlling the magnitude of the step size. Besides this, it introduces several heuristics to stabilize the training and improve the overall performance. Noticeably, it considers  damping the learning rate increase when the function value rises (when compared against previous iterations); and gradually increasing the batch size when some criteria based on function values are met (to reduce the random noise when a local minimum is approached). It demonstrates the effectiveness of the proposed method mostly using vision experiments.

**Strengths:**

- This paper can be followed easily and most heuristics are intuitive. The computation of the step size is cheap.

- The experiments on the 2-dimensional example are interesting and show under certain settings (e.g., rotation) the proposed method can significantly outperform other adaptive step sizes such as Adam and RMSprop.

**Weaknesses:**

- The proposed step size lacks theoretical guarantees even in the convex settings (or even convex quadratics?). I am not sure where the technical challenges are.

- I don’t think the current method is compared fairly with other methods given all the heuristics added on top of the learning rate. It is difficult to tell  where the gain (if there is any) comes from? Is it because of the step size schedule, or batch size increase, or iterates averaging (named as mean value boosting in the paper)? If iterates averaging were applied to other baselines, would the results change?

- Another major concern is that there are many hyperparameters associated with the proposed method, which raises questions regarding its practical usability. How expensive are the tunings of these hyperparameters?

- The experiments on language modelling are inconclusive.

**Questions:**

N/A

---

> ### Author Response · Authors · 2024-11-22
>
> >The proposed step size lacks theoretical guarantees even in the convex settings (or even convex quadratics?). I am not sure where the technical challenges are.
>
> The theoretical challenge lies in the adaptive nature of the learning rate $\alpha$ which is allowed to vary strongly. In such situations it becomes difficult to prove guaranteed convergence. Note that for most convergence proofs it is assumed that $\alpha$ is sufficiently small!
>
>
> >I don’t think the current method is compared fairly with other methods given all the heuristics added on top of the learning rate. It is difficult to tell where the gain (if there is any) comes from? Is it because of the step size schedule, or batch size increase, or iterates averaging (named as mean value boosting in the paper)? If iterates averaging were applied to other baselines, would the results change?
>
> To see, at least to some extent, the influence of the additional heuristics, we have included in the graphics for the accuracy developments the results with and without weight decay and with and without the iterates averaging (see the appendix). Therein, one can see the iterates averaging increase the speed but not the final test-performance (our “boost” results are in the end similar to the results without “boost). Only for CIFAR-10 they do not meet after 200 epochs, but meet after 300 epochs (not visible in the graphic). Note also that the SGD results use weight decay.
>
>
> >Another major concern is that there are many hyperparameters associated with the proposed method, which raises questions regarding its practical usability. How expensive are the tunings of these hyperparameters?
>
> Our intend is to provide an optimizer which does not need tuning of hyperparameters. True, we introduce many new parameters, yet their nature is primarily indirect, i.e. instead of the learning rate $\alpha$ being a hyperparameter, we have hyperparameters which control the update of $\alpha$. Thereby a good $\alpha$ is found by the optimizer during the run, i.e. no pre-scanning is needed. Also our batch size scheduler is intended to work for all/ many networks unchanged.
>
>
> >The experiments on language modeling are inconclusive.
>
> This is true and it might be that ELRA is not competitive on LLMs. However, it we think that it will provide a true gain for networks with limited training sets, such as image processing.

---

### Meta-Review · Area_Chair_76fP · 2024-12-16

**Metareview:**

**Summary:** The authors introduce a new optimizer, ELRA, that uses adaptive step size and adaptive batch, based on the last two gradients. ELRA increases the batch size when the gradient is too noisy. The new algorithm is $\mathcal{O}(n)$ and rotation invariant. They validate their results on several models.

**Strengths:** The method does not require much additional computation, is more robust to the choice of the initial learning rate, and shows some improvements. Rotation invariance is advantageous when rotating the coordinates.

**Weaknesses:** The method lacks theoretical guarantees, even in the convex (quadratic setting), and some of the choices do not have theoretical justification. The method combines several previously known techniques, and it is sometimes unclear where the improvements come from (step size, batch size, averaging, etc.). Some of the assumptions are not theoretically verified (for instance, the closeness of the iterates). Relation to previous methods such as delta-bar-delta and step-size adaptation using exponentiated gradient updates have not been discussed. The method introduces additional hyperparameters, which still require tuning.

**Decision:** Based on the concerns raised by the reviewers, the paper will benefit from 1) additional theoretical results and analysis, 2) discussion and comparison to previous approaches, 3) clean ablations where the effect of each component (step-size, batch-size, averaging, weight decay, etc.) is studied separately. A more thorough benchmarking result based on, e.g., [1], would strongly validate the applicability of the method. I recommend rejection for the current version.

[1] GE Dahl et al. "Benchmarking neural network training algorithms", arXiv preprint arXiv:2306.07179, 2023.

**Additional Comments On Reviewer Discussion:**

The authors clarify some of the misunderstandings by the reviewers (for instance, their method uses a single learning rate instead of a per-parameter one). However, the reviewers are still not sufficiently convinced with the responses.

---

### Decision · Program_Chairs · 2025-01-22

Reject